# Radiation Exposure during Fluoroscopy-Guided Ozone Chemonucleolysis for Lumbar Disc Herniation

**DOI:** 10.3390/jcm11247424

**Published:** 2022-12-14

**Authors:** Matteo Luigi Giuseppe Leoni, Sara Vitali, Fabrizio Micheli, Marco Mercieri, Giustino Varrassi, Roberto Casale, Felice Occhigrossi, Carlo Giordano

**Affiliations:** 1Unit of Interventional and Surgical Pain Management, Guglielmo da Saliceto Hospital, 29121 Piacenza, Italy; 2Department of Medical Physics, Guglielmo da Saliceto Hospital, 29121 Piacenza, Italy; 3Department of Medical and Surgical Science and Translational Medicine, “La Sapienza” University, 00189 Rome, Italy; 4Pain Therapy Unit, Sant’Andrea Hospital, 00189 Rome, Italy; 5Paolo Procacci Foundation, 00193 Rome, Italy; 6Opusmedica Persons, Care & Research-PC&R, 29121 Piacenza, Italy; 7Pain Therapy Department, San Giovanni Addolorata Hospital, 00184 Rome, Italy

**Keywords:** radiation exposure, ozone chemonucleolysis, disc herniation

## Abstract

Introduction: Radiation exposure is a frequent drawback of spinal surgery, even if X-ray guidance plays a pivotal role in improving the accuracy and safety of spinal procedures. Consequently, radiation protection is essential to reduce potential negative biological effects. The aim of this study was to evaluate patients’ radiation exposure, the radiation dose emission during fluoroscopy-guided ozone chemonucleolysis (OCN), and the potential role of patient characteristics. Methods: The radiation dose emission reports were retrospectively evaluated in patients who underwent single-level OCN for lumbar disc herniation. A generalized linear model (GLM) with a gamma distribution and log link function was used to assess the association between radiation emission and patients’ characteristics such as age, sex, BMI, level of disc herniation, disc height, and site of disc herniation. Results: Two hundred and forty OCN cases were analyzed. A safe and low level of radiation exposure was registered during OCN. The median fluoroscopy time for OCN was 26.3 (19.4–35.9) seconds, the median radiation emission dose was 19.3 (13.2–27.3) mGy, and he median kerma area product (KAP) was 0.46 (0.33–0.68) mGy ⋅ m^2^. The resulting KAP values were highly dependent on patient variables. In particular, sex, obesity, and residual disc height < 50% significantly increased the measured KAP, while levels of disc herniations other than L5-S1 reduced the KAP values. Conclusions: The radiation exposure during OCN is low and quite similar to a simple discography. However, patient characteristics are significantly related to radiation exposure and should be carefully evaluated before planning OCN.

## 1. Introduction

Lumbar disc herniation is one of the most common reasons for low back pain, and approximately 80% of the population reports an episode once in their lifetime [1]. Increases in age reduce the proteoglycans production of the disc and leads to dehydration and increased annulus fibrosus strain [2]. These pathological changes result in tears and fissures of the annulus, thus facilitating herniation of the nucleus pulposus. Simultaneously, axial overload of the spine may result in disc material extrusion through a damaged annuls [3]. Posterolateral disc herniations frequently result in nerve root compression and are associated with prominent infiltration of inflammatory cells and proinflammatory cytokine expression in the herniated disc [4]. Consequently, radiating pain occurs in a dermatomal area, and motor or sensory neurological deficits can arise [5]. Conservative treatment is generally recommended. This consists of NSAIDs, muscle relaxants, and/or weak opioids [6], along with physical exercise and rehabilitation [7,8].

If conservative treatments have failed and no progressive neurologic deficits occur, minimally invasive disc surgery should be considered. Several disc procedures are nowadays feasible [9], and ozone chemonucleolysis (OCN) is frequently used with good results [10].

Fluoroscopic imaging guidance plays a pivotal role in improving the accuracy and safety of spinal procedures [11]. Percutaneous disc access is normally performed with the use of fluoroscopy in order to accurately visualize the anatomical structures of the spine and to monitor the trajectory of the needle. However, the use of fluoroscopy exposes both patients and physicians, as well as operating room personnel, to radiation exposure [12]. Radiation protection is a key aspect to improve radiation safety and to reduce potential negative biological effects [13]. Radiation exposure evaluation is another important issue because minimally invasive spinal surgery and interventional spinal procedures are routinely guided by ionizing radiation. If appropriate safety precautions are implemented and maintained, low radiation exposure to patients and physicians is generally found for spinal surgery and interventional procedures [14,15,16].

However, to the best of our knowledge, radiation exposure during OCN in the operating room is a relatively unexplored topic. The aim of this study was to evaluate patients’ radiation exposure and radiation dose emission during fluoroscopy-guided OCN, and whether patient-specific factors may affect the radiation emission necessities.

## 2. Material and Methods

A retrospective database was created with consecutive patients with low back pain and radiculopathy or radicular pain who underwent OCN at the unit of Interventional and Surgical Pain Management, at Guglielmo da Saliceto Hospital, Piacenza, Italy. This study was approved by the local Ethics Committee (Institutional Review Board approval number 570/2022/OSS/ASLPC), and all the patients treated from April 2020 to December 2021 were included. Patients with radicular pain of disco-radicular origin due to disc herniation, documented by magnetic resonance imaging and with no response to conservative treatments, were included. We considered only patients who underwent single-level OCN to reduce the possible bias of frequent fluoroscope adjustments in case of multiple sequential disc herniations.

The OCN technique was extensively described in our previous publication [17]. Briefly, after antibiotic prophylaxis, patients are placed in a prone position, and a conscious sedation is induced. This prevents any possibility of masked accidental nerve root contact, which would not be possible with patients receiving general anesthesia. The skin at the lumbar level is prepared with local antiseptic preparation and a sterile draping. A Chiba-type needle (*20 G, 15 mm, HS Hospital service S.p.A, Aprilia, Italy*) is inserted under oblique intermittent fluoroscopic guidance by the standardized posterolateral extra-articular approach. Both anteroposterior and lateral fluoroscopic views are routinely obtained to confirm that the needle is correctly placed in the center of the intervertebral disc. The physician who performs OCN directly controls the fluoroscopy unit. The discography is then performed and a O_3_-O_2_ 27 mcg/mL mixture is administered into the nucleus pulposus. A periganglionic injection of steroid and local anesthetic with O_3_-O_2_ 27 mcg/mL is subsequently administered.

Patients are normally discharged on the same day of OCN, and the use of a lumbar corset is recommended for the first postoperative week, followed by a gradual resumption of physical activity. All OCN procedures were performed by the same physician with extensive experience in disc surgery.

### 2.1. Radiation Exposure

A C-arm fluoroscope (*GE Healthcare, OEC 9800 Plus*) was used to guide all of the OCN procedures. The fluoroscope had three fields of view (FOV): 31, 23, and 17 cm in diameter. Voltage varied from 40 kV to 120 kV. Radiation was obtained in several ways: continuous fluoroscopy (0.2–10.0 mA normal mode; 0.2–20 mA high definition), pulsed fluoroscopy (0.2–10 mA; 1, 2, 4, 8 pps), high-level pulsed fluoroscopy (0.2–40 mA; 1, 2, 4, 8 pps), digital cine pulse mode (up to 150 mA, 12 pps), digital spot mode (up to 75 mA), and single shot exposure (up to 75 mA). The fluoroscope was used in pulsed fluoroscopy mode with a pulse rate of 8 pulses per second (pps), 50 ms pulse width, and automatic dose adjustment. The distance between X-ray tube focus and patients in the anterior-posterior (A-P) and latero-lateral (L-L) projections were, respectively, 50–55 cm and 45–50 cm. Regular quality controls of both the X-ray tube and the image producer were performed. The following parameters were measured: kerma area product (KAP), (mGy m^2^), air kerma at the interventional reference point (IRP), (mGy), and fluoroscopy time. The IRP was located at 30 cm away from the intensifier entry in the direction of the X-ray tube. The transmission chamber was calibrated with a Piranha multi-X-ray meter (*RTI Group, Flöjelbergsgatan, Sweden*). The calibration factors were 0.8 for the air kerma at the IRP and 1.10 for the KAP. Therefore, the radiation exposure data reported in the tables were corrected for the calibration factors. We decided to use the KAP value as a dependent variable because it gives a more accurate estimation of radiation exposure compared with fluoroscopy time. In fact, if the exposed area is thin, the total dose delivered could be lower even if the fluoroscopy time is longer [18]. Patient radiation exposure data were extracted from the Radiation Dose Structured Report as reported by the C-arm fluoroscope.

### 2.2. Clinical and Demographic Data

Preoperative variables such as age, sex, body mass index (BMI), level of disc herniation, and side of disc herniation (right, left) were collected. BMI was classified according to the World Health Organization (WHO) classification systems as: underweight (BMI < 18.5 kg/m^2^), normal weight (BMI 18.5–24.9 kg/m^2^), overweight (BMI 25–29.9 kg/m^2^), and obese (BMI > 30 kg/m^2^). Disc height at the site of OCN was divided into two categories: residual disc height of the herniated disc ≥ 50% (type A) and residual disc height < 50% (type B) [19,20].

## 3. Statistical Analysis

A descriptive statistic was carried out. Continuous variables are reported as mean ± SD or median and interquartile range (IQR), while categorical data are reported as relative number and percentage. The Shapiro–Wilk test was used for assessing normality. Dichotomous data were compared using the Mann–Whitney U test and polychotomous data using the Kruskal–Wallis test.

Differences between the groups were assessed by the Mann–Whitney U test, and the Dunn’s test with Bonferroni correction was used for the post hoc pairwise comparison of the groups. Therefore, the alpha level for comparison was reduced from 0.05 to 0.01. Linear regression was used to investigate the relationship between the continuous variables KAP and air kerma at the IRP and between KAP and fluoroscopy time.

A generalized linear model (GLM) with a gamma distribution and log link function was used to assess the association of KAP with patients’ age, sex, BMI, level of disc herniation, disc height, and site of disc herniation. Because the gamma regression with logarithmic function had the lowest Akaike information criterion (AIC), it was selected as the appropriate model. GLM is an extension of regression models designed to deal with skewed data with error distributions beyond the normal distribution. To facilitate interpretation, the estimates from the GLM gamma model are presented as the ratio of expected KAP calculated as e^β^, where β is the regression coefficient from the GLM [21]. Finally, the model was compared with the saturated model (ideal model) with the Hosmer–Lemeshow test. By definition, a saturated model is a model that perfectly fits the data.

Variance inflation factors (VIFs) calculation was used to analyze the collinearity between the covariates. Collinearity was excluded because the VIFs were lower than 1.30. Stata MP, version 16.0 (*STATA Corp., College Station, TX, USA*), and R v4.0.3 (*R Foundation for Statistical Computing, Vienna, Austria, www.rproject.org*, accessed on 4 November 2022) were used for the analyses.

## 4. Results

Two patients (<1% of the sample) were removed from the analysis due to missing radiation exposure data. Consequently, a total of 240 patients met the inclusion criteria and were included in the present study. The mean age of the studied population was 53.9 ± 14.4 years and 154 (64.2%) were men. There were 8 (3.3%) underweight, 96 (40%) normal weight, 92 (38.3%) overweight, and 44 (18.4%) obese patients. The median BMI was 25.7 (23.8–28.4 kg/m^2^). A positive correlation was observed between KAP and BMI (r = 0.34, *p* < 0.001), and the median KAP was significantly higher in obese patients (Kruskal–Wallis test, *p* < 0.001).

The majority of OCN were performed at the L5S1 level (48.3%) followed by L4L5 (35.4%) (Table 1). Almost half of the herniated discs (122 discs, 50.8%) were type B (residual disc height < 50%), while the remaining 118 discs (49.2%) were classified as type A (residual disc height > 50%). Left-side disc herniation was the most common (53.8%), while right-side herniation occurred in 111 (46.2%) patients.

The median fluoroscopy time for OCN was 26.3 (19.4–35.9) seconds, while the median air kerma was 19.3 (13.2–27.3) mGy, and median KAP 0.46 (0.33–0.68) mGy ⋅ m^2^. A quite perfect positive linear association was observed, as expected, between air kerma and KAP (r = 0.99, *p* < 0.001, (Figure 1A). A positive linear association was also found between exposure time and KAP (r = 0.70, *p* < 0.001) (Figure 1B).

Fluoroscopy time, KAP, and air kerma at IRP are reported for each disc level in Table 2. A significant difference in KAP values was found between the different levels of disc herniation (*p* = 0.0001). The highest KAP values were observed in L5–S1 discs with a significant difference compared with L4–L5 (*p* < 0.001) and L3-L4 (*p* < 0.001).

As shown in Table 3, all of the variables, apart from age (*p* = 0.07) and the side of disc herniation (*p* = 0.90), had a significant effect on the measured KAP values. The patients’ sex (male) (β = 0.137, OR 1.15, 95% CI 1.12–1.19, *p* = 0.001), BMI > 30 kg/m^2^ (β = 0.14, OR 1.14, 95% CI 1.05–1.25, *p* = 0.002), and disc type B (β = 0.58, OR 1.79, 95% CI 1.60–2.00, *p* < 0.001) had a mathematically positive effect on the KAP values. On the contrary, the level of disc herniation (L2-L3 β = −0.29, OR 0.75, 95% CI 0.58–0.96, *p* = 0.02; L3-L4 β = −0.46, OR 0.63, 95% CI 0.53–0.75, *p* < 0.001 and L4-L5 β = −0.29, OR 0.75, 95% CI 0.67–0.84, *p* < 0.001) had a mathematically negative effect on KAP values, compared with L5-S1 as a reference. The fitted generalized linear model was not significantly different from the saturated model (Hosmer–Lemeshow Test, χ^2^ = 0.66, *p* = 0.99).

## 5. Discussion

Our study demonstrates that KAP is highly dependent on patient variables. In particular, sex, obesity, and residual disc height < 50% significantly increased the measured KAP values. On the contrary, levels of disc herniation different than L5–S1 reduced the KAP values. A positive linear association was observed, as expected, between air kerma and KAP. This result is useful for internal control and to assess data homogeneity. It confirms the absence of a possible collimator effect, because no magnification, no changes in focus spot size, and no increase in the peak of X-ray intensity were used. Radiation protection is an important aspect during minimally invasive spinal surgery. According to the well-known as low as reasonably achievable (ALARA) principle, every effort should be made to minimize patient and operator radiation exposure.

Our data revealed a low radiation exposure during OCN, and the fluoroscopy time was in the low-level range of previously published data for lumbar discography [22]. Unfortunately, it was not possible to compare our data with the diagnostic reference levels (DRLs) because they do not include comparable procedures. However, the radiation exposure of our patients during OCN is quite similar to that of DRLs for fluoroscopically guided facet joint injection and twenty to twenty-five times lower than that of vertebroplasty [23].

Male patients were associated with increased KAP values (OR 1.15, *p* = 0.01) compared with women. Some differences related to sex exist in the lumbar spine and pelvis. Male patients present a higher iliac crest and reduced lordosis and sacral slope compared with female patients [24,25]; the overall cross-sectional areas of vertebral bodies are 25% bigger in men than in women [26].

The quantitative association between BMI and radiation exposure during spinal surgery is a relatively unexplored topic. Sometimes BMI values are simply considered as a binary variable (≥25 kg/m^2^ cut-off), and overweight patients are frequently considered together with obese patients [27]. On the contrary, in accordance with Kukreaja et al. [28], our findings support the idea that only obesity (BMI ≥ 30 kg/m [2]) is associated with increased radiation exposure during minimally invasive spinal surgery. These findings can be explained by fluoroscope-related technical factors. If a fluoroscope is used in a manual mode setting, the radiation exposure is independent of the patient’s size, but the image brightness is adversely affected because a reduced radiation energy reaches the detector in patients with elevated BMI. For this reason, a fluoroscope is generally used in automatic brightness control model, and the device automatically adjusts the quantity of the electron flow and its kinetic energy to increase its penetrating capacity [29].

The level of disc herniation may directly influence radiation exposure during transforaminal lumbar endoscopic spine surgery [30]. In our population, the herniated discs were mainly located at L5–S1 (48.3%) and L4–L5 (35.4%), because approximately 95% of disc herniations in the lumbar area occur at the L4–L5 or L5–S1 levels [1]. We found significantly higher KAP values at the L5–S1 level compared with at the L4–L5 (*p* < 0.001) and L3–L4 (*p* < 0.001) levels. These findings are directly related to the technical difficulty in performing OCN at the L5–S1 level. A fluoroscopy beam caudal adjustment is generally needed in prone patients to visualize the L5–S1 disc space [31], along with a proper identification of the entry point and an accurate selection of the trajectory [32,33]. On the contrary, disc herniations at L4–L5 (OR 0.75, 95% CI 0.67–0.84, *p* < 0.001), L3–L4 (OR 0.63, 95% CI 0.53–0.75, *p* < 0.001), and L2–L3 (OR 0.75, 95% CI 0.58–0.96, *p* = 0.02) had lower KAP values than L5–S1. Disc herniations at L1–L2 were not statistically relevant, as they accounted for only 1.4% of the population sample.

Intervertebral disc degeneration is a progressive morphological, structural, and histological change in the disc [34]. Disc height reduction represents an advanced stage of disc degeneration with significant structural changes with osteophytes and end-plate sclerosis [35]. In our study, a residual disc height < 50% was associated with a significant increase (OR 1.79, *p* < 0.001) of KAP values because, in this case, the capacity of the Chiba needle insertion into the disc was reduced and hardened.

To the best of our knowledge, this is the first study that analyzed the factors associated with radiation exposure during fluoroscopy-guided OCN. In the only previous publication on a similar topic by Somma et al., the authors used fluoroscopy or computed tomography (CT) to guide OCN, and the radiation exposure was reported only as a total value without a detailed analysis based on patient characteristics [36]. Even if Somma et al. recently reported the clinical outcome along with radiation exposure after OCN, some consideration is necessary. First, we did not know patient-specific variables that could have affected the radiation exposure and the choice of using fluoroscopy or CT. As far as we know, there are no data in the literature precisely indicating which among fluoroscopy or CT should be used for OCN. Second, the authors did not specify if they performed a discography while using fluoroscopy for OCN. As we previously reported, disc degeneration grade is strictly related to the outcome after OCN [17]. In our patients, we found a radiation exposure about three times less than that reported by Somma et al. for patients who underwent fluoroscopy-guided OCN. We hypothesize some possible reasons for this finding. First, the settings were different: an interventional radiology suite was used by the cited authors; however, we performed OCN in an operating room with a team highly experienced in spinal surgery and interventional procedures. Second, Somma et al. did not report if the physician who was performing OCN was directly controlling the fluoroscopy unit, or if the physician had supervised the radiographer while performing OCN. This element should also be considered because a different radiation exposure was found in the two different conditions [37]. Finally, two different fluoroscopy devices were used for OCN in our study than by Somma et al.

An additional comparison of our results with radiation exposure during interventional disc surgery was not possible because the existing studies included procedures other than OCN.

In our article, we decided not to report the radiation effective dose. As was previously reported, the effective dose is useful for comparing only different diagnostic examinations, and its evaluation and interpretation are very problematic because organs and tissues receive only partial exposure during interventional procedures [38]. In this context, the effective dose should be interpreted only as a “dosimetric quantity” to optimize radiation protection.

This study has several limitations. In this single-center retrospective study, only one type of C-arm fluoroscope was used. This may limit the extensibility of our results to other fluoroscopy devices. Moreover, the OCN procedures were performed by only a single experienced physician. Although the results may not be extended to evaluate the potential role of different levels of experience in radiation exposure, they may represent a reference standard for further comparative studies. The level of experience could potentially have affected the results because less experience is associated with prolonged radiation exposure [39] and a significant difference was seen in university settings with trainers compared with private practice [40]. It should also be mentioned that in this retrospective large sample size of a population with single-level disc herniation, no particular procedural difficulty or complex anatomy (e.g., scoliosis, spinal deformity, osteoporosis, spondylolisthesis, foraminal stenosis, and vertebral osteophytosis) were reported. This may represent a bias in the estimation of radiation exposure, as complex anatomy could contribute to a longer lasting session and therefore to increased radiation exposure [41].

Finally, although of remarkable importance, the evaluation of physician and assisting personnel’s radiation exposure was beyond the scope of this study.

## 6. Conclusions

Ozone chemonucleolysis (OCN) requires patient radiation exposure through fluoroscopy as do other minimally invasive interventional spinal procedures. We present the first study describing the impact of patient characteristics associated with radiation exposure during fluoroscopy-guided OCN. In particular, sex, obesity, and residual disc height of the herniated disc increased the radiation exposure. On the contrary, levels of disc herniation other than L5-S1 reduced the radiation exposure. All of these anatomical differences should always be considered when performing fluoroscopy-guided OCN as they can influence radiation exposure. If performed by experienced physicians, the radiation exposure is low and quite similar to that of simple discography. However, every effort should be made to avoid unnecessary radiation exposure and to promote safe and effective radiation use.

## Figures and Tables

**Figure 1 jcm-11-07424-f001:**
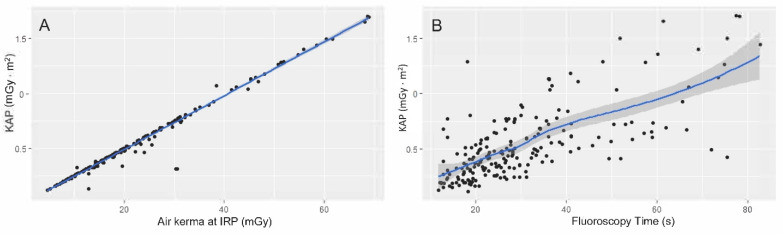
Linear regression of air kerma at IRP with KAP (**A**) and fluoroscopy time with KAP (**B**). A positive linear association was observed in both cases.

**Table 1 jcm-11-07424-t001:** Characteristics of the herniated lumbar discs and radiation parameters.

Variable	Total OCN (*n* = 240) no. (%)
Age, mean ± SD	53.9 ± 14.4
Sex	
Male	154 (64.2%)
Female	86 (35.8%)
BMI	25.7 (23.8–28.4)
<18.5 kg/m^2^	8 (3.3%)
18.5–24.95 kg/m^2^	96 (40%)
25–29.9 kg/m^2^	92 (38.3%)
>30 kg/m^2^	44 (18.4%)
Level of herniated disc	
L1–L2	4 (1.7%)
L2–L3	11 (4.6%)
L3–L4	24 (10%)
L4–L5	85 (35.4%)
L5–S1	116 (48.3%)
Disc height	
Type A (≤50% reduction)	118 (49.2%)
Type B (>50% reduction)	122 (50.8%)
Side of disc herniation	
Left	129 (53.8%)
Right	111 (46.2%)
KAP (mGy · m^2^), median (IQR)	0.46 (0.33–0.68)
Time(s), median (IQR)	26.3 (19.4–35.9)
Dose at IRP (mGy), median (IQR)	19.3 (13.2–27.3)

**Table 2 jcm-11-07424-t002:** Summary of fluoroscopy time, KAP, and air kerma at IRP in different levels of lumbar disc herniation.

Level of Herniated Disc	Fluoroscopy Time (S) Median (IQR)	KAP (mGy m^2^) Median (IQR)	Air Kerma at IRP (mGy) Median (IQR)
L1–L2	24.6 (21.9–30.2)	0.37 (0.29–0.44)	17.8 (15.3–24.2)
L2–L3	25.3 (19.7–32.5)	0.38 (0.17–0.49)	14.3 (9.6–19.6)
L3–L4	18.6 (17.4–30.3)	0.25 (0.20–0.32)	10 (8–13.2)
L4–L5	21.4 (17.1–26.2)	0.43 (0.30–0.60)	17.8 (12–24.2)
L5–S1	30.9 (25.4–41)	0.54 (0.42–0.75)	22.5 (17.1–30.5)

**Table 3 jcm-11-07424-t003:** Factors associated with KAP: results from the GLM.

Level of Herniated Disc	β	OR (95% CI)	*p* Value
Intercept	−1.66	0.19 (0.13–0.28)	0
Age	0.003	1.003 (0.99–1.007)	0.07
Gender (male)	0.137	1.15 (1.03–1.28)	0.01
BMI			
<18.5 kg/m^2^	−0.06	0.94 (0.86–1.04)	0.27
18.5–24.95 kg/m^2^	Ref.	-	-
25–29.9 kg/m^2^	0.04	1.04 (0.99–1.09)	0.10
>30 kg/m^2^	0.14	1.14 (1.05–1.25)	0.002
Level of herniated disc			
L1–L2	−0.40	1.49 (0.42–1.06)	0.09
L2–L3	−0.29	0.75 (0.58–0.96)	0.02
L3–L4	−0.46	0.63 (0.53–0.75)	<0.001
L4–L5	−0.29	0.75 (0.67–0.84)	<0.001
L5–S1	Ref.	-	-
Disc height (type B)	0.58	1.79 (1.60–2.00)	<0.001
Side of disc herniation (right)	−0.0063	0.99 (0.90–1.01)	0.90

## Data Availability

Data are available by the corresponding author, on reasonable request.

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
