# Peer review of "Radiation Exposure during Fluoroscopy-Guided Ozone Chemonucleolysis for Lumbar Disc Herniation"

_jcm, 2022, doi:10.3390/jcm11247424_

Round 1
Reviewer 1 Report
The paper is well written and describe patients factor that may influence radiation dose to patients itself and operators.
Certainly as described by the authors the procedure done a a single experienced operators is a main limits and not depict operators variance that play a main role in interventional procedure.
The authors should describe in details the characteristic of the image intensifier, the distance usually use between X-ray tube and patient both in A-P and L-L if oblique views are used. Finally I understand that standard fluoroscopy was used and not pulsed why please explain.
Author Response
Thank you for these important considerations. The technical characteristics of the fluoroscope along with the distances were added in the manuscript.
The fluoroscope was used in the pulsed fluoroscopy mode with automatic dose adjustment. We clarified it in the text.
Reviewer 2 Report
Reducing medical exposure is an important issue, and for this purpose, multifaceted verification is necessary. This study plays a role in this regard and is a very significant verification of this issue. The paper is also well written.
The following are minor comments.
It should be clearly stated in the objectives and methods that the study targets patient exposure, not occupational exposure. I did not understand that until I read through the discussion.
In addition, the paper indicates that patient exposure in OCN is not large by comparing with reference 22, but I think that is not enough. Why not compare it to the diagnostic reference levels (DRLs)?
In the conclusion, it says "every effort should be made to avoid unnecessary radiation exposure and to promote safe and effective radiation use", which is related to ALARA principle, which is presented by the ICRP in its concept of optimization of protection. I think it would improve the quality of the manuscript if the ALARA principles are also mentioned in the discussion.
The horizontal axis "mGy" in Figure 1(A) is a unit, so the title should be indicated. Also, the unit should be indicated on the vertical axis "KAP".
Throughout, the unit of KAP is written as "mGy*m2," but please confirm that the "*" is an appropriate notation.
Author Response
Reducing medical exposure is an important issue, and for this purpose, multifaceted verification is necessary. This study plays a role in this regard and is a very significant verification of this issue. The paper is also well written.
The following are minor comments.
It should be clearly stated in the objectives and methods that the study targets patient exposure, not occupational exposure. I did not understand that until I read through the discussion.
Thank you very much for this very important suggestion, we clarified it in the text.
In addition, the paper indicates that patient exposure in OCN is not large by comparing with reference 22, but I think that is not enough. Why not compare it to the diagnostic reference levels (DRLs)?
Thank you very much for this suggestion. Unfortunately, it is not possible to compare our data with the diagnostic reference levels (DRLs) since they do not include comparable procedures. However, we added a comparison of our data with facet joint injections and to vertebroplasty, as reported by DRLs. Reference n. 23 was added.
In the conclusion, it says "every effort should be made to avoid unnecessary radiation exposure and to promote safe and effective radiation use", which is related to ALARA principle, which is presented by the ICRP in its concept of optimization of protection. I think it would improve the quality of the manuscript if the ALARA principles are also mentioned in the discussion.
Thank you for the suggestion, ALARA principles were added in the
The horizontal axis "mGy" in Figure 1(A) is a unit, so the title should be indicated. Also, the unit should be indicated on the vertical axis "KAP".
Figure 1 was changed according to your relevant suggestions.
Throughout, the unit of KAP is written as "mGy*m2," but please confirm that the "*" is an appropriate notation.
Thank you for this comment, we replaced the symbol “*” with the symbol “⋅” which is the most used by different Authors.
